# Implementing the Point Spread Function Deconvolution for Better Molecular Characterization of Newly Diagnosed Gliomas: A Dynamic ^18^F-FDOPA PET Radiomics Study

**DOI:** 10.3390/cancers14235765

**Published:** 2022-11-23

**Authors:** Shamimeh Ahrari, Timothée Zaragori, Marie Bros, Julien Oster, Laetitia Imbert, Antoine Verger

**Affiliations:** 1Imagerie Adaptative Diagnostique et Interventionnelle, Institut National de la Santé et de la Recherche Médicale U1254, Université de Lorraine, F-54000 Nancy, France; 2Nancyclotep Imaging Platform, Université de Lorraine, F-54000 Nancy, France; 3Department of Nuclear Medicine, Centre Hospitalier Régional Universitaire de Nancy, F-54000 Nancy, France

**Keywords:** PET, point spread function, machine learning, radiomics, glioma

## Abstract

**Simple Summary:**

The point spread function deconvolution (PSFd), which is known to improve contrast and spatial resolution of brain positron emission tomography (PET) images, has not been evaluated for the routine analysis of amino-acid PET imaging. Our study therefore investigated the effects of applying the PSFd to a radiomics analysis of the clinical dynamic L-3,4-dihydroxy-6-[^18^F]-fluoro-phenyl-alanine (^18^F-FDOPA) PET images (tumor-to-background ratio and time-to-peak parametric images), and evaluated the impact of these effects on the molecular characterization of newly diagnosed gliomas. We show that applying the PSFd to dynamic ^18^F-FDOPA PET images significantly improves the detection of molecular parameters in newly diagnosed gliomas for predicting isocitrate dehydrogenase mutated and/or 1p/19q codeleted gliomas, for a combination of radiomics features extracted from static and dynamic parametric images.

**Abstract:**

Purpose: This study aims to investigate the effects of applying the point spread function deconvolution (PSFd) to the radiomics analysis of dynamic L-3,4-dihydroxy-6-[^18^F]-fluoro-phenyl-alanine (^18^F-FDOPA) positron emission tomography (PET) images, to non-invasively identify isocitrate dehydrogenase (IDH) mutated and/or 1p/19q codeleted gliomas. Methods: Fifty-seven newly diagnosed glioma patients underwent dynamic ^18^F-FDOPA imaging on the same digital PET system. All images were reconstructed with and without PSFd. An L1-penalized (Lasso) logistic regression model, with 5-fold cross-validation and 20 repetitions, was trained with radiomics features extracted from the static tumor-to-background-ratio (TBR) and dynamic time-to-peak (TTP) parametric images, as well as a combination of both. Feature importance was assessed using Shapley additive explanation values. Results: The PSFd significantly modified 95% of TBR, but only 79% of TTP radiomics features. Applying the PSFd significantly improved the ability to identify IDH-mutated and/or 1p/19q codeleted gliomas, compared to PET images not processed with PSFd, with respective areas under the curve of 0.83 versus 0.79 and 0.75 versus 0.68 for a combination of static and dynamic radiomics features (*p* < 0.001). Without the PSFd, four and eight radiomics features contributed to 50% of the model for detecting IDH-mutated and/or 1p/19q codeleted gliomas, respectively. Application of the PSFd reduced this to three and seven contributive radiomics features. Conclusion: Application of the PSFd to dynamic ^18^F-FDOPA PET imaging significantly improves the detection of molecular parameters in newly diagnosed gliomas, most notably by modifying TBR radiomics features.

## 1. Introduction

The point spread function (PSF) correction, including the post-reconstruction PSF deconvolution (PSFd), increases contrast and spatial resolution at the cost of an increased noise level in the reconstruction of positron emission tomography (PET) images by correcting for the system’s depiction of point sources as a function of their location in the field of view [1,2,3]. PSF modelling tested on phantoms showed that the correction was particularly suited to PET radiotracers with focal uptake patterns such as L-3,4-dihydroxy-6-[^18^F]-fluoro-phenyl-alanine (^18^F-FDOPA) [4]. PSFd was also found to be specifically useful for brain PET imaging as it increases the quantitative accuracy of measured activity concentrations [5].

^18^F-FDOPA is a PET amino-acid radiotracer that is clinically relevant for the primary diagnosis of gliomas [6] and whose diagnostic performance improves with the addition of dynamic analysis [7] and the extraction of radiomics features [8]. We previously reported on the diagnostic performance of ^18^F-FDOPA for the molecular characterization of newly diagnosed gliomas in a radiomics study which measured the contributions of dynamic acquisition and radiomics features to identify the molecular parameters of isocitrate dehydrogenase (IDH) mutation and/or 1p/19q codeletion [9]. These molecular parameters are the cornerstone of glioma classification, as indicated in the recent World Health Organization (WHO) glioma classifications [10,11], and are associated with patient prognosis [12]. Our previous study extracted dynamic features using a region-based approach whilst a more recent study evaluated dynamic ^18^F-FDOPA PET imaging based on a voxel-based approach by investigating time-to-peak (TTP) parametric images [13]. The PSFd has, however, not been extensively investigated in the context of amino-acid radiotracers in gliomas [14,15]. The two studies published to date that applied the PSFd only examined its impact on standardized uptake value (SUV)-derived indices but did not investigate impacts on currently used dynamic semi-quantitative parameters or on radiomics features. Moreover, the impact of PSFd modifications of these SUV-derived indices on diagnostic performance achieved in routine clinical practice remains to be evaluated. This is particularly significant since application of the PSFd on reconstructed PET images is not addressed in the current guidelines which define the use of amino-acid PET radiotracers in oncology [16], presumably because no data on the effects of applying such a reconstruction method was available when the guidelines were established.

Our study therefore aims to investigate the effects of applying the PSFd to the radiomics analysis of clinical dynamic ^18^F-FDOPA PET image samples, and to assess whether applying the PSFd improves the ability of specific radiomics derived variables to correctly identify the IDH mutation and/or 1p/19q codeletion status of newly diagnosed gliomas.

## 2. Materials and Methods

### 2.1. Study Population

A total of 57 newly diagnosed glioma patients, who underwent a ^18^F-FDOPA PET acquisition in the Nuclear Medicine Department of the Regional University Hospital (CHRU) of Nancy from February 2018 to February 2021, were selected for our single-center retrospective study. Only patients with (i) a neuropathological diagnosis of grade II–IV glioma, according to the WHO 2021 classification [11], and with (ii) available raw data were included in the analysis. IDH-mutation status was assessed by immunohistochemical detection of IDH1 R132H protein expression (Dianova, clone H09), or by Sanger sequencing for cases which did not stain for IDH1 R132H and also lost the ATRX immunohistochemical signal [17]. Tumors with an oligodendroglial morphology or with an IDH mutation without concomitant loss of ATRX staining were additionally tested for a 1p/19q codeletion using multiplex PCR fragment analysis (loss of heterozygosity), or comparative genomic hybridization [18]. The study was approved by the institutional ethics committee (Comité d’Ethique du CHRU de Nancy) on 26 August 2020. Informed consent was obtained from all study participants. The trial was registered at ClinicalTrials.gov (NCT04469244) and complied with the principles of the Helsinki declaration.

### 2.2. PET Acquisition and Image Reconstruction

All patients fasted for at least 4 h before ^18^F-FDOPA PET acquisition and some patients received Carbidopa, to help increase tracer uptake in the brain, 1 h prior to their exam, which was the institutional procedure effective between February 2018 and September 2020 [19]. Acquisitions were performed on the same digital PET/computed tomography (CT) device (Vereos, Philips Healthcare^®^, Eindhoven, The Netherlands). First, a CT scan was obtained for each patient. A 30 min dynamic PET acquisition was performed following the injection of 2MBq ^18^F-FDOPA per kg of body weight. One static image based on the last 20 min of the acquisition (2 iterations, 10 subsets, 256 × 256 × 164 voxels of 1 × 1 × 1 mm^3^) and 30 frames of 1 min each (3 iterations, 15 subsets, 128 × 128 × 82 voxels of 2 × 2 × 2 mm^3^) for dynamic images were reconstructed using an OSEM 3D algorithm [16,20]. All static and dynamic PET images were reconstructed with and without PSFd. Images were corrected for CT attenuation and dead time, as well as random and scattered coincidences during the reconstruction process.

### 2.3. Segmentation and Image Pre-Processing

Healthy brain, striatum and tumor volumes of interest (VOIs) were manually delimited on the static image by a nuclear physician (M.B.) using the LifeX software (lifexsoft.org, accessed on 6 December 2021) [21]. As recommended in [16], a crescent-shaped VOI was positioned on three consecutive static image slices of the semi-oval center from the unaffected hemisphere so as to include both healthy white and gray matter. Striatum VOIs were delineated semi-automatically using a 70% threshold of the SUV_max_ uptake and tumor VOIs with a threshold of 1.6 of the mean healthy brain SUV (SUV_mean_) [9]. Because of the long acquisition times that may lead to the inclusion of patient movements, dynamic images were registered to CT images for motion correction. To reduce the impact of any potential noise on voxel time activity curves (TACs), dynamic images were also denoised using the highly constrained backprojection method [13,22].

To correct for the impact of Carbidopa on SUV measurements, static images were normalized by the SUV_mean_ of a healthy brain to generate static tumor-to-background-ratio (TBR) parametric images [23]. Voxel/region-based TACs were fitted before being normalized to the fitted mean brain TAC [23] ($TAC_{ratio}=\frac{fitted\;TAC_{tumor}}{fitted\;TAC_{mean\;of\;brain}}$). TTP values were computed as the time interval between the start of dynamic acquisition and the time the maximum uptake value was reached, and the voxel-based analysis enabled the construction of dynamic TTP parametric images [13].

### 2.4. Feature Extraction

Radiomics features were extracted via the open-source pyradiomics Python package (https://github.com/Radiomics/pyradiomics, accessed on 10 June 2022) as well as the image biomarker standardization initiative (IBSI) [24] for local and overall intensity peaks that were not available in the pyradiomics package. One hundred and five radiomics features were extracted from the TBR and TTP individual parametric maps and included first-order statistics, morphological, local intensity, intensity histogram and textural features. To allow a more detailed investigation of the PSFd effect on dynamic parameters, we performed both a voxel/region-based dynamic analysis.

#### 2.4.1. Voxel-Based

Spatial resampling was performed on dynamic TTP parametric images to reach 1 × 1 × 1 mm^3^ isotropic voxels with a linear interpolation, as described by the IBSI [24], and using the SimpleITK Python package [25]. During radiomics features extraction, images were discretized using a fixed bin width of 0.1 SUV for the static TBR and 1 min for dynamic TTP parametric images. Three different feature sets were considered: (i) static (94 radiomics features extracted from static TBR parametric images and 11 morphological features), (ii) dynamic (94 radiomics features extracted from dynamic TTP parametric images and 11 morphological features) and (iii) static/dynamic (94 static TBR radiomics features, 94 TTP radiomics features and 11 morphological features).

#### 2.4.2. Region-Based

Each tumor VOI was defined by 114 features. In addition to extracting radiomics features from static TBR images (105 features), other conventional features like mean, maximum and peak of TBR and tumor-to-striatum ratios, as well as metabolic tumor volume (MTV), were computed (7 features). Region-based dynamic TTP and slope (2 features) parameters were extracted as previously described [9].

### 2.5. Model Training and Validation

All analysis was performed with Python 3.8 using the Scikit-learn library (https://scikit-learn.org/stable/index.html, accessed on 20 June 2022). Classification tasks were divided into two parts: IDH-mutant versus IDH-wildtype glioma, and glioma with versus without 1p/19q codeletion. The range of extracted features was standardized using the z-score method. To reduce the dimensionality of the feature vector, we first removed the zero variance features, then an unsupervised feature selection was performed using complete-linkage feature agglomeration based on the Spearman correlation with a threshold of 0.9 [26]. As the samples for the two classification tasks were not balanced, the Borderline SMOTE process was included in the applied pipeline to oversample the minority class. We trained L1 regularized logistic regression (Lasso) as a proper selection operator that could select useful features and discard redundant ones. To assess model generalizability, a stratified 5-fold cross-validation (CV) with 20 repetitions was applied. For each fold, the model was trained on 80% of samples and validated on an unseen subset of the remaining 20% of samples. The optimal hyperparameter (L1 penalty strength) was selected using a Bayesian search over the train set by minimizing cross-entropy loss and with 300 iterations to further limit random results. The CV performance, which was evaluated using the optimal hyperparameter value, was then used to select the best model and provide an estimate of the expected accuracy of the algorithm. Training and evaluation were performed by deriving different metrics to achieve a more reliable performance.

### 2.6. Statistical Analysis 

Categorical variables were expressed as the number of patients and percentages and continuous variables as medians with interquartile ranges (IQR). Patient characteristics from the primary histopathological groups were compared using the Chi-square test for categorical variables and the Kruskal-Wallis test for continuous variables. To investigate the impact of PSFd on extracted radiomics features, a two-sided Wilcoxon test was applied. Benjamini-Hochberg corrections for multiple comparisons were used [27]. Diagnostic performance (area under the curve (AUC), sensitivity, specificity and balanced accuracy) was computed from model predictions over the test set of the repeated CV folds. One thousand bootstrap iterations of the distribution of the obtained performance were derived and represented as a mean followed by 95% confidence intervals (CI) of individual metrics. Systematic permutation tests were applied to assess whether the models detected a true class structure in the data and performed significantly better than random guessing [28,29]. One-sided Mann-Whitney U tests were applied on the obtained AUCs for the intergroup comparisons between the different feature sets (independent samples) of each individual PSFd status. To measure the superiority of PSFd within each feature set (related paired samples), Wilcoxon tests were also performed. *p*-values smaller than 0.05 were considered statistically significant. All statistical analysis was performed using SciPy 1.7.3. The Shapley additive explanation (SHAP) values were extracted to assess the importance score of features for each classification [30].

## 3. Results

### 3.1. Patient Characteristics

Fifty-seven patients were retrospectively included in the study (median of 56 (IQR, 41–67) years of age, 49% women). Tumor histopathology for the initial diagnosis was either determined from surgery (32%) or biopsy (68%) tissue samples. This allowed to classify the patients into 4 groups, two IDH mutation and two 1p/19q codeletion statuses, that were subsequently used to perform the two binary classification tasks: IDH-mutant (*n* = 24) versus IDH-wildtype (*n* = 33), 1p/19q codeleted (*n* = 12) versus non-codeleted (*n* = 45) gliomas. According to the WHO 2021 glioma classification [11], 12 (21%) cases were IDH-mutant and 1p/19q non-codeleted astrocytomas (66% grade II, 17% grade III and 17% grade IV), 12 (21%) were IDH-mutant and 1p/19q codeleted oligodendrogliomas (100% grade II) and 33 (58%) were IDH-wildtype glioblastomas. Forty-six (81%) patients received Carbidopa premedication before the ^18^F-FDOPA PET acquisition. Further details of the patient characteristics are provided in Table 1. Representative ^18^F-FDOPA PET images with and without the PSFd are shown in Figure 1.

### 3.2. PSFd Impact on Radiomics Features

As detailed in Appendix A, PSFd significantly modified 95% of TBR radiomics features with a median absolute change of 10.14% (IQR, 4.4%–27.1%). This significant change was only observed for 79% of features extracted from dynamic TTP parametric images (with a median absolute change of 3.26% (IQR, 1.7%–6.7%)). It should be noted that no significant modification was observed after applying the PSFd on dynamic parameters from the region-based analysis (*p* = 0.356 after application of the PSFd on the region-based TTP feature, Appendix A). Interestingly, the effects of PSFd were not uniform across all tumors, with a greater impact observed for small lesions with high radiotracer uptake (Figure 2).

### 3.3. IDH Mutation Prediction

Performances for predicting the IDH mutation by the different PSFd statuses and voxel/region-based analysis are presented in Table 2 (performances on train set are available in Appendix A). Applying the PSFd on PET images provided better predictive performance for static and static/dynamic datasets (respectively AUCs of 0.785 (0.756, 0.815) versus 0.686 (0.656, 0.715) and 0.831 (0.804, 0.854) vs. 0.791 (0.765, 0.813), *p* < 0.001). In contrast, there was no difference when the PSFd was applied to the dynamic dataset (*p* = 0.494). Interestingly, applying the PSFd also improved the region-based IDH-mutation prediction (respective AUCs of 0.883 (0.863, 0.903) and 0.827 (0.806, 0.848) with and without PSFd, *p* < 0.001), with a better diagnostic performance compared to the voxel-based analysis (*p* < 0.001). 

In the voxel-based analysis, 4 radiomics features contributed to at least 50% of model performance, but by applying the PSFd, this contribution was reduced to only 3 radiomics features. This reduced the number of TBR parametric image features from 3 to 1 (Figure 3). The region-based TTP was the most important feature for identifying an IDH mutation (61% with PSFd and 44% without) but applying the PSFd reduced the number of contributive TBR parametric features (from 1 feature without PSFd to none with PSFd).

### 3.4. 1p/19q Codeletion Prediction

Performances for predicting the 1p/19q codeletion by the different PSFd statuses and voxel/region-based analysis are presented in Table 3 (performances on train set are available in Appendix A). Applying PSFd on PET images provided better predictive performance for dynamic and static/dynamic datasets (respectively AUCs of 0.721 (0.686, 0.756) versus 0.688 (0.647, 0.727) with *p* = 0.012 and 0.755 (0.725, 0.786) versus 0.683 (0.648, 0.716) with *p* < 0.001). In contrast, a trend was only observed when applying the PSFd to the static dataset (*p* = 0.079). Applying the PSFd improved detection of the 1p/19q codeletion from the region-based analysis (respective AUCs of 0.828 (0.791, 0.860) and 0.787 (0.753, 0.815) with and without PSFd, *p* < 0.001) and resulted in a better diagnostic performance compared to the voxel-based analysis (*p* < 0.001). 

In the voxel-based analysis, 8 radiomics features contributed to at least 50% of model performance, but by applying the PSFd this contribution was reduced to 7 radiomics features (Figure 4). Region-based TTP was the most important feature for identifying the 1p/19q codeletion (29% with PSFd and 19% without), but irrespectively of the application of the PSFd, it contributed less compared to the IDH-mutation prediction (61% with PSFd and 44% without).

## 4. Discussion

The present study demonstrates the effects of applying the PSFd to ^18^F-FDOPA PET images in newly diagnosed gliomas and supports that the application of this correction improves the diagnostic performance for predicting the molecular classification. Interestingly, the effects of PSFd are more pronounced for TBR radiomics features and lead to a better selection of all radiomics features that account for model performance.

To the best of our knowledge, our study is the first to explore the effects of applying PSFd to brain amino-acid PET. However, the fact that 95% of TBR parametric features are significantly modified after application of the PSFd is consistent with neuro-oncology amino-acid PET imaging results reported in literature [14,15]. After applying the PSFd, SUV-derived parameters from our TBR parametric images were significantly modified, with a majority of increased values observed (Appendix A). This may be substantiated by PSFd enhancing contrast ratios [1], particularly for radiotracers with high tumor-to-background ratios, as is the case for amino-acid radiotracers [4]. The PSFd had a greater impact on small-volume tumors with high radiotracer uptake since the partial volume effect is expected to have a greater impact on smaller structures [14]. Girard et al. reported modifications of ^18^F-FDOPA PET kinetic parameters after applying PSFd [15]. We also report modifications of radiomics features derived from TTP parametric images, albeit that these radiomics features are less significantly modified than those from TBR parametric images (Appendix A). We subsequently evaluated modifications induced by the PSFd application on two clinical classification tasks previously shown to be relevant for the initial diagnosis of gliomas with amino-acid PET imaging; namely the prediction of the IDH mutation and the 1p/19q codeletion [9], which are two key-molecular characteristics defined in the current WHO 2021 glioma classification [11].

The PSFd increased the voxel-based diagnostic performance of IDH-mutation prediction with AUCs for the static/dynamic dataset of 0.83 with PSFd versus 0.79 without (*p* < 0.001 for AUC comparison, Table 2). This improvement in diagnostic performance was also observed for the static dataset (Table 2). Interestingly, without PSFd, the dynamic information outperforms that obtained from static images for predicting the IDH mutation, which is consistent with our previous findings of overlapping but different patient populations [9,18]. Indeed, the AUC provided by the static/dynamic dataset from the region-based analysis was similar to that previously reported (0.88 versus 0.83 [9]), with TTP being the main predictive feature with or without PSFd (Figure 3). Application of the PSFd induces modifications of radiomics features that allow for a better selection of relevant radiomics features from TBR parametric images (from 3 features without PSFd to 1 after PSFd).

The PSFd also enhanced the voxel-based diagnostic performance of 1p/19q codeletion prediction, with AUCs for the static/dynamic dataset of 0.75 with PSFd versus 0.68 without (*p* < 0.001 for AUC comparisons, Table 3). Significant improvement was also observed for the dynamic dataset (*p* = 0.012 for AUC comparisons with and without PSFd status, Table 3) with a trend only observed in the static dataset (*p* = 0.079, Table 3). Although relatively less modified by application of the PSFd than radiomics features from TBR parametric images, radiomics features from TTP parametric images are also significantly modified (please see Appendix A). As shown in Figure 4, for the region-based analysis, TTP remains an important relevant value to identify the 1p/19q codeletion with or without PSFd, but to a lower extent than observed for prediction of the IDH mutation. The SHAP values again suggest that applying the PSFd reduces the number of radiomics features that contribute to the model (Figure 4). To further investigate the effects of PSFd on static TBR and dynamic TTP parameter images, the model was also fed a combination of radiomics features extracted from static TBR with PSFd and radiomics features derived from dynamic TTP without PSFd (as performed in our previous publication [9]). No significant differences were observed for either classification tasks compared to the combination of static and dynamic feature sets, both reconstructed with PSFd, although a non-significant trend was observed for predicting the 1p/19q codeletion (respectively 0.819 (0.793, 0.841) versus 0.831 (0.804, 0.854), *p* = 0.226 and 0.717 (0.681, 0.751) versus 0.755 (0.725, 0.786), *p* = 0.061 for IDH mutation and 1p/19q codeletion).

Interestingly, predictive performances obtained from the region-based analysis were better than those from the voxel-based analysis (Table 2 and Table 3, *p* < 0.001). This is in contrast to previous studies that evaluated the predictive value of amino-acid PET imaging for the TERT mutation, where the voxel-based analysis of dynamic images was able to identify the mutation whilst the region-based analysis was not [31,32]. This important point highlights the fact that the detailed information provided by the voxel-based analysis is not always efficient, particularly when region-based parameters are more robust at predicting a particular classification, as is the case for the TTP-based prediction of the IDH mutation [9,18]. In this case, radiomics analysis of TTP parametric images does not provide the same information as region based TTP, which is designed to capture the TTP of the most aggressive part of the tumor. Dynamic image analysis in current clinical routine practice is based on region-based parameters, which is supported by our results indicating that dynamic region-based parameter performances are better than those from the voxel-based analysis. However, as presented in Appendix A, PSFd significantly modified extracted radiomics features and had a stronger impact on TBR features. However, there was no significant effect of PSFd on region based TTP (*p* = 0.356, Appendix A). The amplification of noise resulting from the application of the PSFd on dynamic images in the region-based analysis would, furthermore, lead to a deterioration of the dynamic information. Application of the PSFd did, however, result in a more accurate selection of TBR radiomics features and a better predictive performance of the model as well. For a region-based analysis in routine clinical practice we would therefore suggest that the PSFd be solely applied to extract information from static PET images.

The current study has several limitations. Firstly, it involved a relatively low number of patients (*n* = 57), although the number of patients included in the current study is in line with case numbers analyzed in the neuro-oncology radiomics PET imaging literature [8]. It should also be emphasized that the primary objective of the current study was to investigate whether or not the PSFd modifies the texture of PET images (radiomics analysis), rather than to provide a predictive model directly. The current study also uses a robust methodological approach adapted to the small number of cases studied. We included standardized features, applied an unsupervised feature selection, balanced the classes and then employed a simple and linear Lasso-like model with an L1 penalty as a regularization operator and used a stratified 5-fold CV with 20 repetitions to assess a more generalized model. A permutation test was also applied to ensure that the models were able to detect the true class structure in the data. Secondly, the PSFd applied after the reconstruction process is an approach that is specifically tailored to our PET system, which restricts the generalization of our results. Nevertheless, applying the PSFd across different systems reveals the same image parameter modifications [1,4,5]. The denoising process to generate voxel-based TACs in the current study could beneficially limit the noise caused by the PSFd, but also reduce the effects of the PSFd on dynamic images. However, as previously reported, limiting noise is critical when extracting voxel-based TACs [5,13], and improving the dynamic dataset for predicting 1p/19q codeletion is at odds with eliminating PSFd effects in the dynamic dataset. The effects of implementing the PSFd have only been validated for a single clinical indication and their scope in terms of other common indications, such as detecting glioma recurrences, remains to be investigated. Given that the detection of glioma relapses is primarily based on static parameters [13,33], any contrast improvement of parametric TBR images, achieved by application of the PSFd, is also expected to improve diagnostic performance. Finally, the current study specifically examined supra-centimetric tumors, thereby reducing the risk of any potential quantitation errors arising from application of the PSFd [34].

## 5. Conclusions

To summarize, the application of the PSFd to ^18^F-FDOPA PET imaging of newly diagnosed gliomas enhance the predictive performance of the molecular characterization of gliomas by increasing the contrast ratio. This primarily impacts parametric TBR imaging and enhances any potential uptake differences observed in gliomas imaged by amino acid PET, which is an imaging modality characterized by a high tumor-to-background ratio.

## Figures and Tables

**Figure 1 cancers-14-05765-f001:**
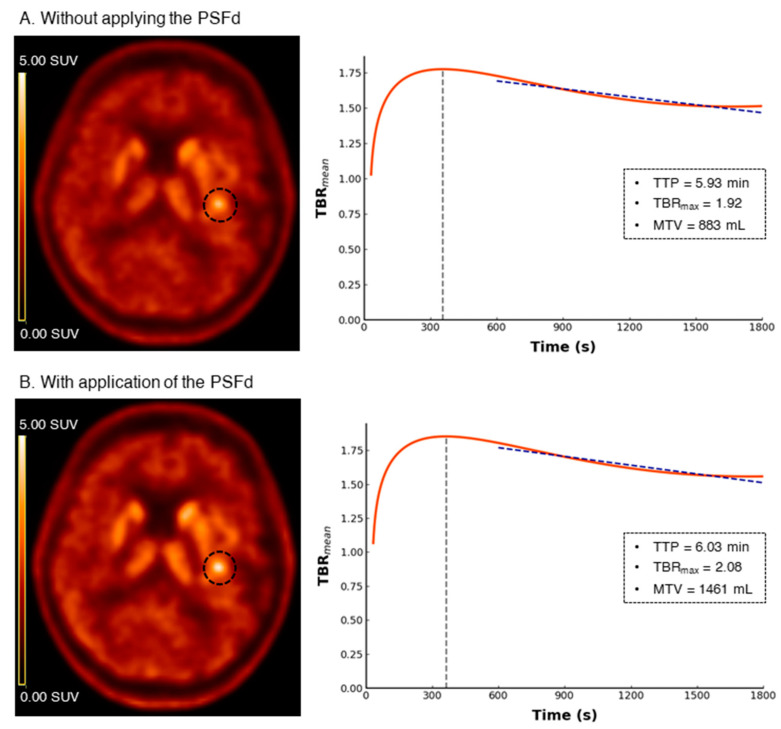
Representative ^18^F-FDOPA PET images for without applying the PSFd (**A**) and with application of the PSFd (**B**) alongside the related region-based time-activity-curve ratios (TAC_ratio_). Tumor volumes of interest are demonstrated on PET images with dashed circles. Application of the PSFd increases the contrast in tumor uptake, but effects on the TAC_ratio_, and consequently effects on TTP, obtained from the region-based analysis are limited. Patient characteristics: 75-year-old female patient classified as an IDH-wildtype glioblastoma. PSFd: point spread function deconvolution; TTP: time-to-peak; SUV: standardized uptake value; TBR_mean_: mean of tumor-to-background-ratio; TBR_max_: maximum of tumor-to-background-ratio; MTV: metabolic tumor volume.

**Figure 2 cancers-14-05765-f002:**
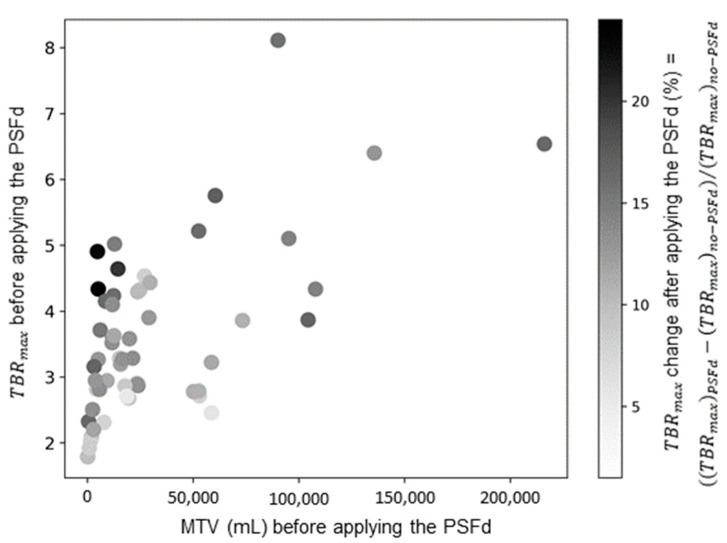
Distribution of TBR_max_ changes after applying the PSFd. TBR_max_: maximum of tumor-to-background-ratio; MTV: metabolic tumor volume; PSFd: point spread function deconvolution.

**Figure 3 cancers-14-05765-f003:**
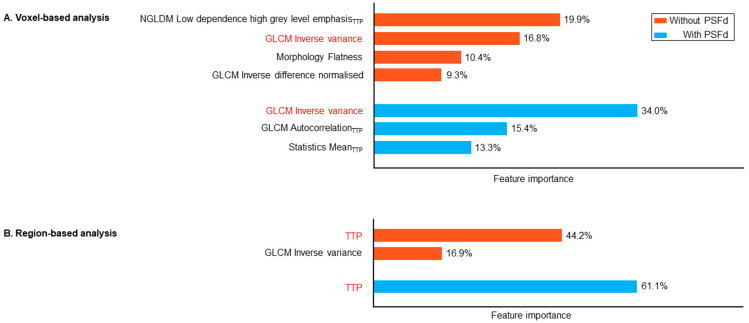
Shapley additive explanation (SHAP) values representing feature importance to substantiate at least 50% of the IDH-mutation prediction model (for voxel-based (**A**) and region-based (**B**) analysis) trained on the static/dynamic feature set. Common features between the two PSFd statuses are highlighted in red. Subscript TTP indicates that the relevant feature is extracted from the dynamic TTP parametric map. PSFd: point spread function deconvolution; TTP: time-to-peak; NGLDM: neighboring grey level dependence matrix; GLCM: grey level co-occurrence matrix.

**Figure 4 cancers-14-05765-f004:**
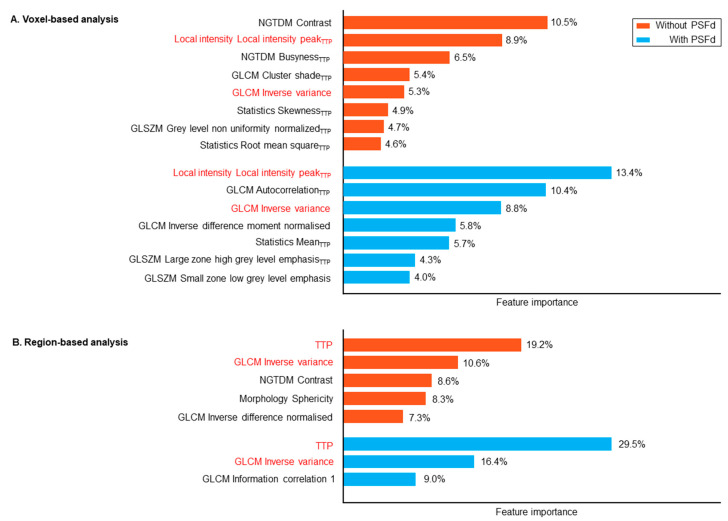
Shapley additive explanation (SHAP) values representing feature importance to substantiate at least 50% of the 1p/19q codeletion prediction model (for voxel-based (**A**) and region-based (**B**) analysis) trained on the static/dynamic feature set. Common features between the two PSFd statuses are highlighted in red. Subscript TTP indicates that the relevant feature is extracted from the dynamic TTP parametric map. PSFd: point spread function deconvolution; TTP: time-to-peak; NGTDM: neighborhood grey tone difference matrix; GLCM: grey level co-occurrence matrix; GLSZM: grey level size zone matrix.

**Table 1 cancers-14-05765-t001:** Patient Characteristics.

	Astrocytoma IDH-Mutant and 1p/19q Non-Codeleted	Oligodendroglioma IDH-Mutant and 1p/19q Codeleted	Glioblastoma IDH-Wildtype	*p*
	*N* = 12	*N* = 12	*N* = 33	
Age, *median (IQR)*	41 (27–57)	48 (41–62)	62 (54–71)	0.003 ^#^
Sex, *n* (%)				
Female	7 (58)	6 (50)	15 (45)	0.745
Male	5 (42)	6 (50)	18 (55)	
Tumor resection, *n* (%)				
Surgery	7 (58)	6 (50)	5 (15)	0.007 ^#^
Biopsy	5 (42)	6 (50)	28 (85)	
Histopathological WHO grade, *n* (%)				
Grade II	8 (66)	12 (100)	-	<0.001 ^#^
Grade III	2 (17)	-	-	
Grade IV	2 (17)	-	33 (100)	
Carbidopa premedication, *n* (%)	10 (83)	11 (92)	25 (76)	0.473
TBR_mean_ *, *median (IQR)*	1.92 (1.8–2.0)	1.95 (1.8–2.1)	2.12 (1.9–2.3)	0.023 ^#^
TBR_max_ *, *median (IQR)*	2.84 (2.5–4.0)	2.83 (2.6–3.6)	3.58 (2.9–4.4)	0.068

* From PET images without PSFd; ^#^
*p*-value is significant for the comparison of the three types of tumor histopathology. IDH: isocitrate dehydrogenase; TBR_mean_: mean of tumor-to-background-ratio; TBR_max_: maximum of tumor-to-background-ratio; MTV: metabolic tumor volume; WHO: World Health Organization.

**Table 2 cancers-14-05765-t002:** Model performance on the test set for IDH-mutation prediction.

	Without PSFd				With PSFd			
Features/Metrics	AUC	Sensitivity	Specificity	B_ACC	AUC	Sensitivity	Specificity	B_ACC
	**Voxel-based analysis**							
Static	0.686 ^ξ^ (0.656, 0.715)	0.797 (0.760, 0.831)	0.496 (0.458, 0.532)	0.686 (0.656, 0.715)	0.785 *^,ξ^ (0.756, 0.815)	0.869 (0.838, 0.897)	0.542 (0.509, 0.572)	0.706 (0.684, 0.728)
Dynamic	0.759 ^¥,ξ^ (0.730, 0.787)	0.733 (0.697, 0.771)	0.640 (0.605, 0.675)	0.686 (0.661, 0.710)	0.764 ^ξ^ (0.737, 0.791)	0.743 (0.709, 0.777)	0.653 (0.622, 0.686)	0.698 (0.676, 0.719)
Static/Dynamic	0.791 ^¥,§,ξ^ (0.765, 0.813)	0.755 (0.720, 0.791)	0.630 (0.593, 0.666)	0.693 (0.666, 0.718)	0.831 *^,¥,§,ξ^ (0.804, 0.854)	0.810 (0.777, 0.843)	0.672 (0.636, 0.707)	0.741 (0.719, 0.763)
	**Region-based analysis**							
Static/Dynamic	0.827 ^‡,ξ^ (0.806, 0.848)	0.667 (0.620, 0.718)	0.760 (0.727, 0.792)	0.714 (0.689, 0.739)	0.883 *^,‡,ξ^ (0.863, 0.903)	0.666 (0.626, 0.709)	0.858 (0.828, 0.887)	0.762 (0.740, 0.785)

*p*-values relate to AUC comparisons. * *p*-value significant for the same feature sets compared to without PSFd status; ^¥^ *p*-value significant for the same PSFd status compared to static feature sets; ^§^ *p*-value significant for the same PSFd status compared to dynamic feature sets; ^‡^ *p*-value significant for the same PSFd status compared to the voxel-based analysis; ^ξ^*p*-value significant compared to permutation test. AUC: areas under the curve; B_ACC: balanced accuracy; PSFd: point spread function deconvolution.

**Table 3 cancers-14-05765-t003:** Model performance on the test set for 1p/19q codeletion prediction.

	Without PSFd				With PSFd			
Features/Metrics	AUC	Sensitivity	Specificity	B_ACC	AUC	Sensitivity	Specificity	B_ACC
	**Voxel-based analysis**							
Static	0.664 ^ξ^ (0.633, 0.693)	0.564 (0.505, 0.627)	0.604 (0.569, 0.636)	0.584 (0.551, 0.617)	0.681 ^ξ^ (0.652, 0.710)	0.552 (0.492, 0.607)	0.623 (0.594, 0.656)	0.588 (0.556, 0.618)
Dynamic	0.688 ^ξ^ (0.647, 0.727)	0.617 (0.555, 0.678)	0.695 (0.663, 0.728)	0.656 (0.624, 0.688)	0.721 *^,¥,ξ^ (0.686, 0.756)	0.628 (0.567, 0.692)	0.686 (0.656, 0.716)	0.657 (0.624, 0.690)
Static/Dynamic	0.683 ^ξ^ (0.648, 0.716)	0.650 (0.595, 0.708)	0.688 (0.653, 0.718)	0.669 (0.636, 0.702)	0.755 *^,¥,ξ^ (0.725, 0.786)	0.590 (0.532, 0.653)	0.728 (0.697, 0.759)	0.659 (0.624, 0.691)
	**Region-based analysis**							
Static/Dynamic	0.787 ^‡,ξ^ (0.753, 0.815)	0.679 (0.615, 0.733)	0.726 (0.698, 0.754)	0.703 (0.672, 0.729)	0.828 *^,‡,ξ^ (0.791, 0.860)	0.750 (0.692, 0.802)	0.790 (0.760, 0.820)	0.770 (0.741, 0.797)

*p*-values relate to AUC comparisons. * *p*-value significant for same feature sets compared to without PSFd status; ^¥^*p*-value significant for same PSFd status compared to static feature sets; ^‡^ *p*-value significant for same PSFd status compared to the voxel-based analysis; ^ξ^*p*-value significant compared to permutation test. AUC: areas under the curve; B_ACC: balanced accuracy; PSFd: point spread function deconvolution.

## Data Availability

Available in Appendix A.

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
