# Peer review of "Implementing the Point Spread Function Deconvolution for Better Molecular Characterization of Newly Diagnosed Gliomas: A Dynamic ^18^F-FDOPA PET Radiomics Study"

_cancers, 2022, doi:10.3390/cancers14235765_

Round 1

Reviewer 1 Report

The authors used FDOPA PET imaging with/without point spread function deconvolution (PSFd) to generate radiomic data and explored the diagnostic ability of predicting IDH1 mutant (n = 24) and 1p/19q co-deletion (n = 12; all also had IDH1 mutation) vs. the other 33 glioma patients without these genetic alterations. The authors showed that PSFd improved the model performance of using radiomics to predict IDH1 and 1p/19q co-del. This is a very interesting paper with significant findings. 

Section 2.1. How were the 1p/19q and IDH1 status determined? Please speciy. 

Section 3.1. It would be nice to show a Table to compare the clinical characteristics among IDH1 mutant, 1p/19q co-deletion, and others. The characteristics to be compared may include age, gender, tumor grading, tumor size, tumor site, SUVmax, etc. with Fisher's exact or Kruskal-Wallis P values. A Venn diagram may also help. 

Section 3.4. In the predictive model, who compares with whom is important. Were patients with IDH1 mutant but without 1p/19q co-deletion (n = 12) used in the model to predict 1p/19q? Were these IDH1-mutant, 1p/19q non-co-del patients (n = 12) considered the same group with or a different group vs. IDH1-WT, 1p/19q non-co-del patients (n = 33)? 

Since 1p/19q co-deleted tumors usualy also confer IDH1 mutation (n = 12 of 12), would they share some radiomic features with 1p/19q non-co-del tumors with IDH1 mutation (n = 12) 

Tables 1 and 2. Because it is a relatively small dataset, I suggest that the case numbers used to calculate the (i.e. the numerators and denominators) be added into the Tables.  If the tables become too length, at least the denominators should be listed in the footnote or text. 

For the broad readership of this journal, I suggest that the authors add in sample FDOPA PET images to explain the effect of PSFd enhancement in predicting IDH1 mutation and 1p/19q co-del. 

In the Discussion Section, please also discuss how the nucear medicine physicians may apply PSFd and the prediction models into clinical PET imaging and reports in the future. 

Reviewer 2 Report

From an epidemiology/biostats point of view, I have some questions for the Authors:

1- Radiomics is nowadays quite fashionable: IMHO, nothing more. ML/AI algorithms need thousands and thousands of stats units to be estimated. What message do 57 pts convey to us? How these conclusions are affordable? How can you prove that this research is not underpowered?

2- line 32, report all p-values with 3-sign digits, never using </>0.05

3- there's no table I, main pts characteristics

4- line 89, levodopa premedication was not standardized, why?

5- line 123, 105 radiomics features for 57 pts? can you prove your model was not deeply overparametrized? how?

6- line 164, biostats section lacks several infos (what about descriptive/inferential approach, i.e.?)(independent or paired data?)

7- line 181 and everywhere, continuous covariates need to be reported only as median(IQR), without using mean/sd

8- how can you prove you results could be generalized? it does not seem so

Reviewer 3 Report

Some areas of improvement.

1. Full meaning of the acronym should be included in the abstract and figure legends.

2. Introduction should have more information on IDH mutation and 1p/19q codeletion.

3. I am not sure how IDH mutation or 1p/19q codeletion was detected in the first place and how the model was validated after that. The results and methods sections should have that information.

4. Patient demographics/cancer status should be included.

Reviewer 4 Report

Dear Author,

Your manuscript is well written and Results are well supported by data and conclusion.

Round 2

Reviewer 1 Report

The authors have addressed all review questions thoroughly with appropriate edits. 

Reviewer 2 Report

My previous concerns do remain unsolved, mainly the sample size which is totally unsuitable for radiomics reliable results